# The Role of Eye Tracking Technology in Assessing Older Driver Safety

**DOI:** 10.3390/geriatrics5020036

**Published:** 2020-06-07

**Authors:** David B. Carr, Prateek Grover

**Affiliations:** 1Department of Medicine and Neurology, Washington University School of Medicine, St Louis, MO 63110, USA; 2Department of Neurology, Washington University School of Medicine, St Louis, MO 63110, USA; groverp@wustl.edu

**Keywords:** driving, rehabilitation, eye tracking, older adults, Alzheimer’s disease (AD)

## Abstract

A growing body of literature is focused on the use of eye tracking (ET) technology to understand the association between objective visual parameters and higher order brain processes such as cognition. One of the settings where this principle has found practical utility is in the area of driving safety. Methods: We reviewed the literature to identify the changes in ET parameters with older adults and neurodegenerative disease. Results: This narrative review provides a brief overview of oculomotor system anatomy and physiology, defines common eye movements and tracking variables that are typically studied, explains the most common methods of eye tracking measurements during driving in simulation and in naturalistic settings, and examines the association of impairment in ET parameters with advanced age and neurodegenerative disease. Conclusion: ET technology is becoming less expensive, more portable, easier to use, and readily applicable in a variety of clinical settings. Older adults and especially those with neurodegenerative disease may have impairments in visual search parameters, placing them at risk for motor vehicle crashes. Advanced driver assessment systems are becoming more ubiquitous in newer cars and may significantly reduce crashes related to impaired visual search, distraction, and/or fatigue.

## 1. Introduction

### 1.1. Objective

A growing body of literature focuses on the use of eye tracking (ET) technology to understand the association between objective visual parameters and higher order brain processes such as cognition. One of the settings where this principle has found practical utility is in the area of driving safety. The purposes of this manuscript are: (a) to review the visual system neuroanatomy and physiology; (b) to describe the clinical bedside exam of eye movements; (c) to describe methodologies to capture eye movements using current technology; (d) to summarize current findings on eye tracking parameters, aging and neurological disease; and (e) to review studies to date on eye tracking and driving. It is hoped that this overview will be useful for both clinicians and scientists as we continue to unravel the complex relationships between aging, neurological disease, driving safety, and countermeasures to decrease crash risk.

### 1.2. Methods

A review of the entire literature on eye tracking, aging, and neurological disease is beyond the scope of this paper. However, we summarize many aspects of these topics to provide context and background for a more focused review on driving and eye tracking in older adults. Neither a systematic review nor a meta-analysis was attempted since there have been very few ET and older adult driving studies. Ovid Medline 1946- and Embase.com 1947- were searched in January 2020. In total, 113 articles were retrieved in Medline and 59 were retrieved in Embase. Inclusion criteria were any eye tracking studies that focused on driving and included a sample of older adults. Of the entire Medline and Embase records, 5 studies were identified that studied ET and older adult drivers. We also identified 4 additional studies when reviewing the reference sections of the papers that were identified in the search. The rest of the records were excluded since they were not eye tracking studies and/or they did not include older adult driver samples. Thus, 9 articles are the subject of our review under the older driver subsection entitled “Older Drivers and Eye tracking”.

The search strategies were designed using a combination of keywords and controlled vocabulary to describe eye movement measurements and driving outcomes such as simulation or traffic accidents regardless of age so to be inclusive. A fully reproducible search strategy used for Ovid Medline is as follows:“eye tracking”/exp/mj OR ((eye OR visual OR optical) NEAR/1 (track*)):ti,ab,de,kw AND (“car driving”/exp OR “traffic accident”/exp OR (automobile* OR car OR cars OR vehicle*):ti OR ((driv*) NEAR/1 (license* OR simulation*)):ti,ab OR ((driving* OR drive*) NEAR/3 (automobile* OR car OR cars* OR vehicle*)):ti,ab OR ((accident* OR collision*) NEAR/2 (traffic* OR road* OR streetcar* OR automobile* OR car OR vehicle OR motorc* OR vehicular)):ti,ab OR ((road) NEAR/2 (test*)):ti,ab OR ((motor) NEAR/1 (vehicle*) NEAR/1 (crash* OR accident* OR collision*)):ti,ab)

## 2. Background

### 2.1. Visual System Anatomy

The neuroanatomy of eye movements is described in Figure 1 [1]. There are four muscles that attach to the eye to direct movement and assist gaze. They are innervated by three cranial nerves in the brainstem, cranial nerves III, IV, and VI [2]. The medial and lateral rectus control horizontal movements and the superior and inferior rectus assist vertical movements. The sympathetic nervous system controls the dilator pupillae muscle which dilates the pupil while the parasympathetic fibers innervate the contractor pupillae muscle and contract the pupil. The extent of activation of each of these systems determines pupil diameter [3]. Two functional classes of eye movements are gaze stabilization and gaze shifting. The former stabilizes the eyes when the head moves, whereas the latter stabilizes the image on the fovea as the target moves. The vestibulo-ocular reflex adjusts eye movements to compensate for head movements [4] and the opto-kinetic reflexes help to avoid image drift on the retina [5].

### 2.2. Eye Movements

Two common eye movements that can be tracked include fixation and saccades. Fixation refers to visual gaze being maintained in a single location for period of time. Saccades are rapid movements that occur for a period from one fixation target to another [6] and both the amplitude and direction of this movement can be measured. These movements are initiated in the frontal lobe and the posterior parietal cortex assists with gaze corrections. Pursuit (smooth pursuit) refers to the eyes following a moving object [7]. The temporal lobe along with the occipital lobe appear to initiate these movements. Vergence occurs when the eyes accommodate to sharpen acuity using the fovea when focusing on near or distant targets. Pupil dilation can occur, is easily measured, and is associated with adaptation [8]. Some of these anatomic pathways are presented in Figure 1 (permission pending).

### 2.3. Bedside Examination

A detailed visual system examination requires knowledge, skill, and equipment. Tests for examining the afferent visual system includes visual acuity, color vision, visual field testing, and ophthalmoscopy. Pupillary examination is integral in identifying the multiple syndromes described in the literature. For the purpose of this article, the main focus is ocular motility. Although eye tracking by computers can give sophisticated and detailed parameters, the clinician is able to do an assessment of these variables in the office setting. Vergence can be assessed by moving an object close to the face. Saccades can be tested by either verbally-guided or self-paced methods [9]. Self-paced saccades can be observed by putting out a finger to the left and right of central fixation and asking the patient to gaze back and forth between the two stimuli. The patient is asked to alternatively look at the examiner’s nose and finger or to find a stimuli that suddenly appears (pen light) in the periphery. The clinician can observe for any delays in initiation (e.g., optic apraxia), dysconjugate gaze (e.g., cranial nerve lesion), decreased range of motion and improvement with smooth pursuit (e.g., progressive supranuclear palsy), reduced speed (e.g., Huntington’s Disease), accuracy (e.g., hyper- or hypometria with cerebellar disease), or saccadic intrusions (e.g., opsoclonus from paraneoplastic syndrome). When testing smooth pursuit, the examiner can request the patient to follow a finger or penlight that is about 40 cm from the midpoint and in an H pattern. This will test all extraocular muscles and appropriate fields [10]. In the smooth pursuit exams, impairments may indicate lesions in the cortex, a cogwheeling jumpy pursuit with parietal lobe disease, and slowness or reduced speed associated with myriad neurodegenerative diseases such as Parkinson’s disease, progressive supranuclear palsy, multisystem atrophy, and cortico-basal ganglionic degeneration [11]. In the anti-saccade test, a stimulus is presented to one side of the patient and the patient is asked to gaze in the opposite direction. The presence of the inability to inhibit a reflex saccade indicates an abnormal response. Anti-saccades are very specific for AD, but are present in other neurodegenerative disorders.

### 2.4. Tracking Methods

Eye tracking can be described as a technique to capture eye position over time during the performance of tasks. There are various methodologies to capture eye tracking variables. Oculometry, or the study of eye movements, has been captured by either electrooculographic (EOG), galvanometric, or corneal reflective techniques. The EOG measurement requires use of electrodes placed around the eyes. As the eyes move from midpoint to the periphery, the retina approaches one electrode while the cornea approaches another one, causing a change in electric field or a dipole which can be measured. The EOG typically has filters to reduce background noise. Vertical and horizontal directions can be assessed. Thus, saccadic eye movement with gaze shifts can be readily detected, but gaze direction and smooth pursuits are not so effectively measured. The head needs to be immobilized and thus it can be effectively used for computer or simulator studies that do not require head movements. There is also a galvanometric technique that can measure changes in electromagnetic fields. This is accomplished by contact lenses that are place on the cornea. The lenses are typically fitted with a magnetic field sensor or an embedded mirror allowing for eye movements in all directions. However, it is not possible to use this in naturalistic settings, and it is basically restricted to past research laboratory efforts [12]. Another common ET method of choice uses infra-red light that reflects off the cornea and will follow the center of the pupil [13]. The point of regard (POR) is a method of calibration that requires the eyes position relative to the head to be stable. The head typically is positioned in a chinrest. Calibration can be challenging for this equipment when trial sessions are long, contact or corrective lenses are used, or there are long eye lashes [14]. Modern videography and computer algorithms can measure parameters using reflections from eye layers and appearance based techniques using facial feature recognition [15]. These are discussed in more detail in the section on Advanced Driving Assistance Systems (ADAS).

### 2.5. Eye Tracking Parameters

Typical ET videography gathers many data which need to be reduced and/or analyzed. While there is a plethora of eye tracking parameters, many common ones are defined with reference to the specified region of data collection, referred to as the “Area of interest (AOI)”. “Revisit” implies gaze returning to an AOI. “Spatial density and transitions” refer to position distribution within AOI and eye movements in and out of AOIs, respectively, and are indicative of visual search strategy and scanning. These scan paths capture positional changes of eye movements over time and can be described as saccade–fixation cycles. “Heat maps” are visual tools representing distribution of eye movements using color intensity. hot zones along with scan paths depict areas where the participant has focused gaze at a high frequency and consequently reveal blind areas which were relatively ignored. These visual representations of eye tracking frequency are typically used in business settings to determine responses to marketing materials and what type of presentation is likely to attract a consumer’s attention.

“Position duration measures” describes length of time for an eye movement, such as “fixation duration”. Fixation within an AOI is defined as “First fixation duration (FFD)”. “Gaze duration” or “Dwell time” is the sum of all fixations and saccades within one “AOI visit”. “Frequency measures” include the number of times an eye movement is observed within a trial, such as “fixation count” and “saccade count”. An indirect measure of fatigue is known as “PERCLOS,” which refers to the percentage of eye closure over the pupil, and has also been studied in the context of driving [16,17].

### 2.6. Eye Tracking Vendors

Since many industries use eye tracking technology, there are multiple vendors that make available equipment at varying levels of cost, complexity, and application. A list of common manufacturers, ranked by descending number of publications (as mentioned on: https://imotions.com/blog/top-eye-tracking-hardware-companies/, 3 March 2020) is included below along with other vendors.

Order Based on Publications:Tobii, https://www.tobii.com/SensoMotoric Instrument, acquired by Apple, no active websiteEyeLink., https://www.sr-research.com/Smart Eye, https://smarteye.se/research-instruments/se-pro/LC Technologies, https://eyegaze.com/category/assistive-tech/page/5/

Additional Vendors:6.Gazepoint, https://www.gazept.com7.ITU Gaze Tracker is a video-based open source tracker: It is hosted through SourceForge: https://sourceforge.net/projects/gazetrackinglib/.8.Cogain Association: https://www.cogain.org9.Attention Tool by iMotions Eye Tracking Solutions: http://www.imotionsglobal.com/10.Interactive Minds: https://www.interactive-minds.com/eye-tracking, http://www.interactive-minds.com/en/eye-tracking-software

We do not endorse any specific company, but provide this information for those interested in embarking on this expanding area of research.

## 3. Eye Movements, Aging and Neurodegeneration

### 3.1. Eye Tracking, Aging and Cognitive Impairment

Episodic or anterograde memory is often impaired early in Alzheimer’s disease (AD). ET technology can be used in the visual paired comparison task (VPC), which taps into this cognitive construct and has been associated with hippocampal injury [18]. In adults, in both healthy and clinical populations, the VPC task as administered by ET was found to be a good measure of recognition memory with the potential to predict normal adults who will convert to MCI [19] and patients with MCI who will convert to AD [20]. Impaired eye tracking parameters have been associated with executive function tests such as Trailmaking Tests [21]. Pupil dilation is controlled by the brain’s locus-ceruleus, which depends on the norepinephrine system. This system controls attention and arousal and has been associated with task difficulty and mental effort. Eye blinking rates are associated with dopamine levels and are increased in learning tasks, working memory, and decision making [22]. Differences between young and older adults in regards to eye tracking and attentional focus appear to diminish when participants can choose their own stimulus [23]. Older adults have also been noted to have more post-saccadic oscillations compared younger adults, which may have an impact on visual perception and measuring trajectories [24]. These studies when recruiting older adults do not typically rate the presence or absence of dementia based on sensitive clinical interviews and have not obtained biomarkers of Alzheimer’s disease. Thus, it is difficult to know if these differences are due to advanced age or the presence of very early neurodegenerative disease.

### 3.2. Eye Tracking and Alzheimer’s Disease

The literature on eye tracking and Alzheimer’s disease includes studies that describe how eye movements differ in neurodegenerative disease or how they relate to impairments in cognitive function [25]. Compared to controls, participants with AD often have poorly regulated gaze patterns and gaze perseveration [26]. Increased blink occurrence and impaired eye-head coordination have been noted [27]. Smooth pursuit has also been found to decline in AD participants with decreases in velocity and saccadic intrusions [28]. The bulk of eye-tracking studies would suggest that anti-saccades are most specific for AD [29]. There appears to be a loss of attentional reserve in studies of visual search indicating excessive saccades and fixations, along with pupil dilatation [30]. AD participants have more trouble with inhibition and go/no go situations [31], and they also are more distracted during smooth pursuit [32]. Pupillary amplitude and velocity dilatation have been noted to be decreased in AD participants compared to controls [33,34]. Even young onset Alzheimer’s disease has been studied with ET and higher-order visuoperceptual impairments have been identified, suggesting these parameters might even be biomarkers for clinical trial outcomes [35].

### 3.3. Eye Tracking Examinations and other Neurodegenerative Disease

Frontotemporal Dementia (FTD) has been studied in regards to eye tracking parameters and their association with cognitive domains. Spatial anticipation impairment has been noted in the behavioral variant of Frontotemporal Dementia (FTDbv) [36]. In FTD, primary progressive aphasia ET has been effective in determining non-verbal responses to stimuli by studying pupil diameter [37], as well as in separating out different language variants, such as primary progressive aphasia vs. semantic dementia [38]. Amyotrophic Lateral Sclerosis (ALS), which can accompany FTD in some cases, has been studied and visual search and executive function task impairment have been detected [39]. Cognitive screening ET tests have been developed for this condition [40]. Smooth pursuit and saccades have been shown to be impaired in Parkinson’s disease (PD) patients depending on the severity [41]. The authors noted that ET studies can often take away the confounding motor skill impairment that traditional psychometric tests have difficulty in avoiding. One of the classic findings in PD is hypometric saccades [42]. Cognitive workload has been noted to be impaired in PD patients [43] along with attentional skill sets [44]. Corticobasal ganglionic degeneration has been noted to have slowed saccadic velocity and delayed initiation [9]. Progressive supranuclear palsy has also been shown to have slow saccadic velocity and hypometric saccades [45], while a study of multisystem atrophy showed square wave jerks or saccadic intrusions [46].

### 3.4. Eye Tracking and Neurorehabilitation

Eye tracking technology has diagnostic as well as therapeutic applications for Neurorehabilitation [47]. Individuals with quadriplegia often have very limited upper extremity movement, and eye tracking is helpful in facilitating independence with communication as well as mobility with the use of environmental control units. Neglect has been characterized in patients with stroke using eye tracking technology [48], which in turn has been helpful in understanding the relationship with functional progress. Stroke patients have also been studied with ET during completion of Trailmaking Tests, indicating impaired visual tracking systems [49]. In a small study, brain injury patients were evaluated in a driving simulator task with ET and were found to still be competent in their skills [50]. Eye tracking technology is also being investigated to identify concussion in sports related injuries [51]. It is useful as an examination tool for subtle eye findings in multiple sclerosis [52] and has applications for understanding visuo-motor coordination in cerebral palsy [53]. Participants with glaucoma [54] or visual field loss [55] were found to have delayed hazard response, opening the doors for possible visual driving rehabilitation strategies in these patients.

## 4. Eye Tracking and Driving

### 4.1. Visual Search, Eye Tracking and Driving

Eye tracking methods have been used to study driving using simulators in older adults (Figure 2).

“Glance-monitoring technologies” refer to eye tracking in the real world of driving [57] where the location and duration of fixation can be measured. The location can be used for determining where information is being processed since information can be in front of the vehicle, to the sides, and behind or within the vehicle (attention maintenance). Hazard anticipation can be measured by determining whether the driver checks key locations in the field of view. Decreasing attention maintenance [58] and hazard anticipation [59] have both been associated with crash risks. The technologies available in vehicles include head-mounted eye trackers, head-mounted scene trackers, vehicle-mounted eye trackers, vehicle-mounted scene trackers, and a combination of modalities [57]. In situations where both eye and scene trackers are used, the eye position can be captured as they fixate on a particular aspect of the scene, aided by specialized software. Head-mounted eye trackers do not require installment but may be uncomfortable to wear and possibly distracting during actual driving. Both vehicle- and head-mounted eye trackers require calibration. In general, saccades in driving last a duration of 60–200 ms and glance duration/fixations approximately 0.1–2 s [12]. Face and/or head camera movements can be scored by a video review and do no not require calibration. Simulator data suggest that, for every 25% increase in duration of a glance, there will be a cost of 0.39 s in brake reaction time and 0.06 s in the standard deviation of lateral position [60].

### 4.2. Inexperienced and Experienced Drivers

Inexperienced young drivers typically have elevated crash rates compared to middle-aged drivers, especially during the first three years of licensure. Inexperience in youth is not uncommonly cited as a cause for a motor vehicle crash. One of the common citations for crashes in the UK and US is “failure to look properly” or “looked but did not see” [61]. Inexperienced drivers have been noted to have a reduced horizontal visual search [62]. Early eye tracking studies suggested that experienced drivers may process stimuli in the environment more efficiently, spend less time with fixations, and also focus on areas in the visual field that are most likely to present potential hazards [63]. However, more recent studies cast doubt on major differences in visual search characteristics between inexperienced and experienced drivers. A recent meta-analysis of high quality studies did not find differences between fixations and vertical search strategies, but did find smaller horizontal search strategies with inexperience drivers [61]. This has obvious implications for inexperienced driver training in mitigating crash risk from peripheral stimuli.

Distracted driving is a growing problem and is believed to significantly contribute to crashes in both novice and experienced drivers [64]. Activities that are distracting can be classified as visual (glancing at a billboard), physical (hands off the steering wheel), cognitive (focusing on other tasks), or auditory (conversations). Examples of distractions include reading, watching TV, eating, calling, texting, etc. One can measure the time of a glance away from the front roadway or the total time away from frontward gaze.

Distraction, fatigue, and aggressive driving behavior may contribute to over 90% of road crashes [65]. Eye tracking devices have been used to provide support for the phenomenon of highway hypnosis revealing drowsiness and impaired eye tracking performance with prolonged drives [66]. Although PERCLOS has been studied in laboratory conditions, it is often not practical in real driving situations since the pupil is impacted by brightness. The use of PERCLOS in vehicles is still limited because of factors such as brightness of on-coming vehicles, which can affect pupil size and accurate measurements. Recently, investigators have used a new algorithm examining saccadic intrusions as an estimate of the mental workload while driving and predicting unsafe driving situations [67].

### 4.3. Older Drivers and Eye Tracking

Older drivers (age ≥65 years) may be less accurate in scene identification, especially at night [68]. Older drivers may have impairment in hazard anticipation, having been noted to check less for traffic during right- and left-hand turns [69]. This is consistent with simulator data that note decreased scanning behavior in older adults [70,71]. Optical blur has been noted to reduce the number of optical fixations and durations along with smaller saccades in older adult drivers [72]. However, this is not a consistent finding, and one study found no decrements in hazard detection with older adults [73]. Older drivers compared to glaucoma counterparts were noted to have larger saccades but both groups showed impaired motion sensitivity [54]. Older drivers with better executive function were shown to have more frequent gaze fixations when moving into roundabouts and through intersections [74]. Direct feedback to older drivers of their taped video trips along with educational simulated driving sessions doubled the number of secondary looks for hazard detection, indicating potential for educational interventions with crash reduction [75]. Older adults who volunteer for these types of studies may not have been rigorously checked for pre-clinical AD or Mild Cognitive Impairment (MCI). Cross section and longitudinal studies have shown driving errors in older adults with pre-clinical AD [76,77]. Further studies are needed to determine the role of very early neurodegenerative disease in deteriorating driving behaviors that may put older adults at risk.

It should be noted that, just because a driver gazes toward a certain area, it does not necessarily imply that the information has been cognitively processed. Older adults have been cited through the years for “looked but did not see” (LBDNS). This phenomenon is reported in the literature more often in older adult drivers [78]. One would expect LBDNS to increase as distractions (e.g., conversations and cell phone) and cognitive load becomes higher. Vehicles that are less conspicuous or salient (e.g., motorcycles) [79], pedestrians in more complex urban environments [80], auditory stimuli [81], and even electronic billboards [82] can result in late fixation during eye tracking driving tasks, indicating possible increase risk in crash risk. Experiments that require the participant to respond to the stimuli (e.g., stop sign or traffic light) or verbally require them to describe past scenery capture cognitive processing of these objects [83].

### 4.4. Limitations of Studies

Challenges of this type of eye tracking car technology includes calibration issues, use of eye glasses, and glare [12]. Sunny days, compared to those with cloud cover, could yield different results and can be particularly challenging to manage. Head-mounted cameras may be distracting or uncomfortable, whereas vehicle-mounted cameras require installation. Drivers may demonstrate variability in regard to the degree their eyes are open and eyelids can occlude the eye [84]. There are always limitations in accurately determining what the participant is looking at or processing, even when eye tracking technology can tell us specifically where the gaze is fixated.

## 5. New Directions

### Eye Tracking in Advanced Driving Assistance Systems (ADAS)

Due to the frequency of crashes, there is a growing desire to reduce visual scanning errors and their causes. It is estimated that motor vehicle crashes account for over 1,000,000 deaths and 20,000,000 injuries per year globally, with costs that may represent 2% of a country’s GNP [15]. A significant reduction in crashes using facial feature recognition with ADAS technology is a major goal of many manufacturers. The models of detection used in these situations are myriad and beyond the scope of this paper. In brief, technologies can use the shape, feature, and/or appearance of the eyes; capture eye blinks, eye, and/or head motion; or adapt some type of combination. The objective is to identify and track the gaze locations and direction, also known as the point of regard (POR). To adequately assess these parameters, eye tracking paradigms typically assess both orientation and position of the head along with location of the eye. Trackers can also measure the percent road center (PCR) or the time fixations fall into a specific area of the road during a determined time period. It has been shown the PCR will increase with cognitive or mental demand [85]. The visual algorithms of these technologies obtain input to determine the drivers gaze and area focus and thus can be a measure of inattentiveness. Visual or auditory alarms will sound when a driver does not achieve minimal pre-set requirements. These systems have been adopted in many vehicles and may use PERCLOS, head nodding, blink rates or speed, gaze direction, head orientation, eye closure, or adapt some combination of these. Costs, calibration limitations, and lack of efficacy limit their widespread application for the public. However, similar to cameras that provide a field of view when backing up, they are likely to become standard as technologies improve and costs are reduced.

## 6. Conclusions

Visual eye tracking and gaze parameters are operated by a very complex and intricate set of muscles, nerves, and neurological systems with input from the cortex to the brainstem. Clinicians can grossly examine important eye tracking variables such as saccades and smooth pursuit, which can assist with the diagnosis of common neurodegenerative diseases. ET technology is becoming less expensive, more portable, easier to use, and readily applicable in a variety of clinical settings. Initial studies that focused on driving and ET used simulators, but now equipment allows for the accurate study of driving behavior in naturalistic settings. This capability will allow for more extensive observations in the field and likely assist with fitness to drive evaluations. This has already been taking place with inexpensive video capture technology that is triggered by G-forces [86]. However, the massive amounts of data require reduction and analysis and can be unwieldy. Older adults with neurodegenerative disease may have impairments in visual search putting them at risk for motor vehicle crashes. Older adults, and especially those with preclinical or presymptomatic Alzheimer’s disease, may have decrements in eye tracking parameters. Studies should be considered for examining ET parameters with AD biomarkers. Preliminary studies suggest that driver education has the potential to improve visual search strategies. Advanced driver assessment systems are becoming more ubiquitous in newer cars and may significantly reduce crashes related to impaired visual search, distraction, and/or fatigue.

## Figures and Tables

**Figure 1 geriatrics-05-00036-f001:**
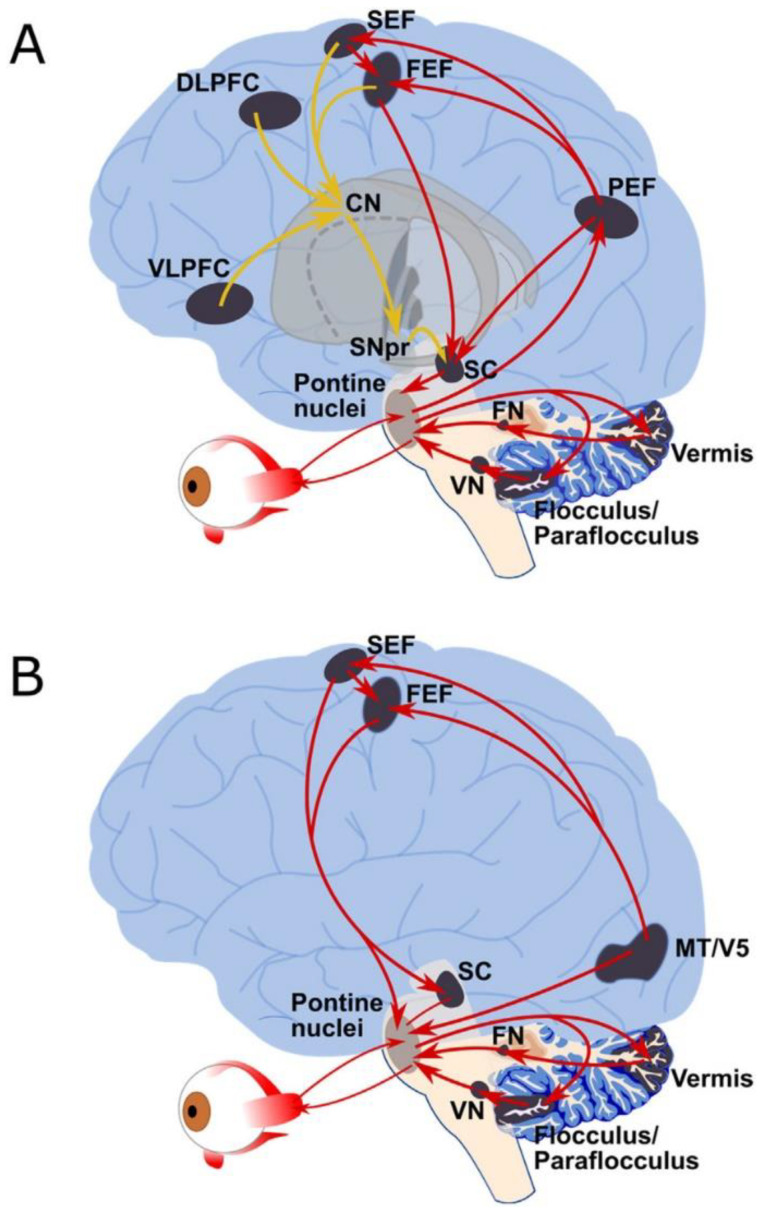
Neuroanatomy of saccadic and pursuit eye movements. Overview of the human ocular motor networks, detailing the descending pathways that control (**A**) saccadic, and (**B**) pursuit eye movements. Grey shaded regions indicate key ocular motor areas, while arrows indicate anatomical connections (may not be direct connections in all cases). (**A**) Saccade pathways: red arrows indicate the *direct pathway* (PEF, FEF, SEF) to SC and brainstem premotor regions, while yellow lines indicate the *indirect pathway* to the SC and brainstem premotor regions via the basal ganglia (striatum, subthalamic nucleus, globus pallidus and substantia nigra pars reticula. (**B**) Pursuit pathways: red arrows indicate the main pathways that control pursuit eye movements. CN = caudate nucleus (basal ganglia); DLPFC: dorsolateral prefrontal cortex; FEF = frontal eye fields; FN = fastigial nucleus, MT/V5 = middle temporal area; PEF = parietal eye fields; pontine nuclei = premotor nuclei (paramedian pontine reticular formation (PPRF) and rostral and caudal interstitial fasciculus nucleus of the medial longitudinal fasciculus (riMLF, cMRF); SC = superior colliculus; SEF = supplementary eye field; SNpr = substantia nigra pars reticulate; Vermis = cerebellar vermis lobules VI and VII; VLPFC = ventrolateral prefrontal cortex; VN = vestibular nuclei. (For interpretation of the references to colour in this figure legend, the reader is referred to the web version of this article.) Figure and text with permission from: Johnson B.P., Lum J.A., Rinehart N.J., Fielding J., Ocular motor disturbances in autism spectrum disorders: Systematic review and comprehensive meta-analysis, Neurosci Biobehav Rev. 2016 Oct; 69, 260–279.

**Figure 2 geriatrics-05-00036-f002:**
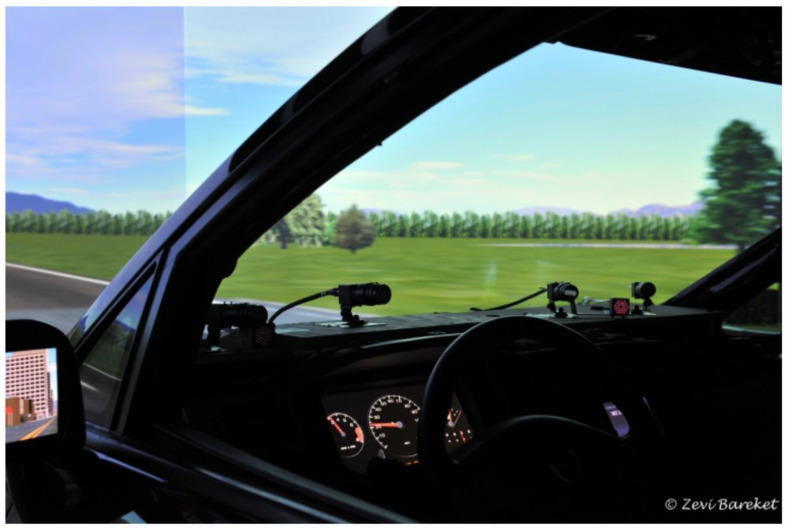
A SmartEye four-camera eye-tracking system in a virtual driving simulator environment (http://www.umtri.umich.edu/what-we-offer/driving-simulator, accessed on 18 March 2020, with permission given from University of Michigan Transportation Institute (UMTRI) [56].

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
