# Peer review of "The Role of Eye Tracking Technology in Assessing Older Driver Safety"

_geriatrics, 2020, doi:10.3390/geriatrics5020036_

Round 1

Reviewer 1 Report

This is an important and timely narrative review looking at eye movements, how to track them and some early understanding of how these change with aging and neurodegenerative disease. This is then related to driving and future car safety systems. It is thorough and covers most of the information I would look for in such a review.

I am recommending a couple of major changes and a number of minor changes.

MAJOR

  • Overall the paper is well set up, however, I would rename the section beginning line 153: EYE TRACKING IN AGING AND COGNITIVE IMPAIRMENT and start that section with more information about how aging in the absence of neurodegeneration affects eye movements.
  • I would prefer to have the methodology for the review, which currently show up half way through the paper, early on in the paper.
  • I would start the paper with an introduction: which sets out the objective, the methodology for the review (see number 2) and how the paper is organized: 1) Intro (objective and methodology) 2) Background (Neuroanatomy of eyes, examination and devices), 3) Eye movements Aging and Neuro-Degeneration 4) impact on driving and 5) Conclusion

MINOR

Line 11: “is focuses”

Line 14: remove “have”

Line 16: “anatomy and physiology”

Line 33: you use “dilator” so to be consistent use “contractor” in line 34

Line 34: “which dilates the pupil…”

Line 38: replace “eye” with “image?”

Line 40: insert “avoid image drift?”

Line 43: remove first “eye”

Line 44: “for a period…”

Lines 70-72: rewrite: something like: “ask the patient to alternatively look at the examiner’s nose and finger, or to find…”

Lines 79-80: rewrite: “in the smooth pursuit exam, ____ impairments may suggest lesions in the cortex”

Somewhere in this section (c. Bedside Exam) include how to do the anti-saccade test – it is referred to later

Line 82: “Parkinson’s disease,…”

Line 89: “Oculometry, or the study of eye movements, has been…”

Line 92: “periphery, the…”

Line 93: “one, causing…”

Line 98: “galvanometric technique that can measure…”

Line 103: there appears to be something missing between “…for” and “uses…”

Line 105: “eye’s position relative to the head to be stable.”

Lines 126-127: “PERCLOS, which refers to the percentage of time the eye lid is covering the pupil, and has been…”

Line 132: “and the type of presentation likely to…”

Line 140: for clarity I would label the first 5 as the “Top 5;” and the other 5 I would label as “Others” and not number them

Line 157: “impaired hippocampal injury.” Pick one.

Line 158: “clinical populations, the…”

Line 162: “brain’s”

Line 170: “how they relate to…”

Line 171: “Compared to controls, participants…”

Line 179: Are both pupillary amplitude and velocity in dilatation decreased? Please rearrange sentence.

Line 190: “variants, such as”

Line 191: “ALS, which can accompany FTD in some cases, has”

Line 192: “have been detected”  

Line 223 (hopefully moving to first section in the paper): rewrite: something like: “Neither a systemic review nor a meta-analysis were possible since there were very few ET studies…. We queried Ovid Medline from 1946-…”

Line 239: “and driving, with special…”

Line 250: refer

Lines 252 – 253: I assume what is going on behind the vehicle is also important.

Line 255: “Decreases in attention maintenance and….”

Line 263: “…approximately 0.1 – 2 seconds…”

Line 266: 0.39 & 0.06

Line 271: when first operating a motor vehicle,

Line 281: inexperienced

Line 291: hazard detection, indicating

Lines 297-298: rewrite something like “…laboratory conditions, however its applicability for use in vehicles is still limited because of factors such as brightness of on-coming vehicles which can affect pupil size and hence measurement of PERCLOS.”

Line 340: “Due to the frequency of crashes, there is a growing desire….”

Line 346: “facial monitoring” or “facial feature recognition”

Line 349: “…or adapt some combination of these.”

Line 350: “direction, also known as…”

Line 356: “and thus can be a measure…”

Line 360: “Costs, calibration limitations, and lack of evidence of efficacy….”

Line 376: rewrite

Line 377: “adults, and especially… disease, may have…”

Line 380: “…are due to advancing age alone or if they are a bio-marker of neuro-degeneration.”

Check the following references for missing names or formatting issues: 3, 9, 10, 46, 47, 64

Author Response

Ms. Frankie Yang, M.Sc.

Editor, MDPI

Room 2207, Jincheng Center, No. 21 Cuijingbeili, Tongzhou

District, Beijing 101121, China

Geriatrics Editorial Office

E-mail: geriatrics@mdpi.com

Manuscript ID: geriatrics-781345

Type of manuscript: Review

Title: The Role of Eye Tracking Technology in Assessing Older Driver Safety

Authors: David Carr *, Prateek Grover

Received: 6 April 2020

E-mails: dcarr@wustl.edu, groverp@wustl.edu

Submitted to section: Geriatric Neurology,

Aging and Driving: 2019

May 17, 2020

Dear Ms. Yang,

We have had the opportunity to review the excellent suggestions by the reviewers and have made all of their suggested changes without exception. We believe the manuscript is better organized and strengthened. We look forward to another review.

Sincerely,

David Carr

REVIEW #1: MAJOR

  • Overall the paper is well set up, however, I would rename the section beginning line 153: EYE TRACKING IN AGING AND COGNITIVE IMPAIRMENT and start that section with more information about how aging in the absence of neurodegeneration affects eye movements.

Response: We agree. We changed the subsection title as you suggested from previous line 153 to new line: EYE TRACKING AND AGING and COGNITIVE IMPAIRMENT. We also changed the section title  which can now be found on new line XXX (previous line 153) from EYE TRACKING AND COGNITIVE IMPAIRMENT TO EYE MOVEMENTS, AGING AND NEURODEGENERATION as recommended below. We believe this title better encompasses the subsections that follow. We have also added additional studies on aging and eye tracking. However, the reviewers will acknowledge that it is often quite difficult to exclude neurodegeneration from aging unless one uses very sensitive clinical tools such as the Clinical Dementia Rating (CDR) and/or the use of PET/CSF Alzheimer’s biomarkers. We did not find any studies with this methodology. These limitations are touched upon later in the manuscript.

  • I would prefer to have the methodology for the review, which currently show up half way through the paper, early on in the paper.

Response: We agree and have moved this up in the paper as also mentioned in the next reviewer #1 comment. However, a comprehensive review on all areas of eye tracking, aging, and neurological disease is beyond the scope of this paper. We have mentioned this statement in the new Methods section.

  • I would start the paper with an introduction: which sets out the objective, the methodology for the review (see number 2) and how the paper is organized: 1) Intro (objective and methodology) 2) Background (Neuroanatomy of eyes, examination and devices), 3) Eye movements Aging and Neuro-Degeneration 4) impact on driving and 5) Conclusion

Response: We agree and appreciate this suggestion from the reviewer. We believe this change significantly strengthens the organization of our paper. We have re-organized the manuscript using the titles for these specific sections.

REVIEWER #1 MINOR

Sincere apologize to this reviewer for all the punctuation and grammar changes that needed to be made throughout the manuscript. They were not only quite necessary but also quite appreciated. In the future, we will take Frankie up on the English review recommendation.

Line 11: “is focuses”

Changed.

Line 14: remove “have”

Removed

Line 16: “anatomy and physiology”

Changed.

Line 33: you use “dilator” so to be consistent use “contractor” in line 34

Changed.

Line 34: “which dilates the pupil…”

Changed.

Line 38: replace “eye” with “image?”

Changed.

Line 40: insert “avoid image drift?”

Changed.

Line 43: remove first “eye”

Removed

Line 44: “for a period…”

Added.

Lines 70-72: rewrite: something like: “ask the patient to alternatively look at the examiner’s nose and finger, or to find…”

Changed.

Lines 79-80: rewrite: “in the smooth pursuit exam, ____ impairments may suggest lesions in the cortex”

Done.

Somewhere in this section (c. Bedside Exam) include how to do the anti-saccade test – it is referred to later

Added.

Line 82: “Parkinson’s disease,…”

Changed.

Line 89: “Oculometry, or the study of eye movements, has been…”

Changed.

Line 92: “periphery, the…”

Changed.

Line 93: “one, causing…”

Changed.

Line 98: “galvanometric technique that can measure…”

Changed.

Line 103: there appears to be something missing between “…for” and “uses…”

Modified.

Line 105: “eye’s position relative to the head to be stable.”

Changed.

Lines 126-127: “PERCLOS, which refers to the percentage of time the eye lid is covering the pupil, and has been…”

Changed.

Line 132: “and the type of presentation likely to…”

Changed.

Line 140: for clarity I would label the first 5 as the “Top 5;” and the other 5 I would label as “Others” and not number them

Changed.

Line 157: “impaired hippocampal injury.” Pick one.

Deleted impaired.

Line 158: “clinical populations, the…”

Changed.

Line 162: “brain’s”

Changed.

Line 170: “how they relate to…”

Changed.

Line 171: “Compared to controls, participants…”

Changed.

Line 179: Are both pupillary amplitude and velocity in dilatation decreased? Please rearrange sentence.

Yes. Clarified.

Line 190: “variants, such as”

Changed.

Line 191: “ALS, which can accompany FTD in some cases, has”

Changed.

Line 192: “have been detected”  

Changed.

Line 223 (hopefully moving to first section in the paper): rewrite: something like: “Neither a systemic review nor a meta-analysis were possible since there were very few ET studies…. We queried Ovid Medline from 1946-…”

We overhauled this area in the new Methods section which is now at the beginning of the paper.

Line 239: “and driving, with special…”

Changed.

Line 250: refer

Changed.

Lines 252 – 253: I assume what is going on behind the vehicle is also important.

Agree. This sentence has been modified.

Line 255: “Decreases in attention maintenance and….”

Changed.

Line 263: “…approximately 0.1 – 2 seconds…”

Changed.

Line 266: 0.39 & 0.06

Changed.

Line 271: when first operating a motor vehicle,

Deleted.

Line 281: inexperienced

Changed.

Line 291: hazard detection, indicating

Changed.

Lines 297-298: rewrite something like “…laboratory conditions, however its applicability for use in vehicles is still limited because of factors such as brightness of on-coming vehicles which can affect pupil size and hence measurement of PERCLOS.”

Modified.

Line 340: “Due to the frequency of crashes, there is a growing desire….”

Changed.

Line 346: “facial monitoring” or “facial feature recognition”

Changed.

Line 349: “…or adapt some combination of these.”

Changed

Line 350: “direction, also known as…”

Changed.

Line 356: “and thus can be a measure…”

Changed.

Line 360: “Costs, calibration limitations, and lack of evidence of efficacy….”

Changed.

Line 376: rewrite

Done.

Line 377: “adults, and especially… disease, may have…”

Changed.

Line 380: “…are due to advancing age alone or if they are a bio-marker of neuro-degeneration.”

Done.

Check the following references for missing names or formatting issues: 3, 9, 10, 46, 47, 64

References have been corrected.

Reviewer 2 Report

Review for Geriatrics

The Role of Eye Tracking Technologies in Assessing Older Driver Safety

This manuscript has a detailed background – good (and unpretentious) description of the visual system and eye movements.  It would be more interesting if they also connect this information with driving.  The other sections are also interesting, albeit it might be too long. Again, when possible, it would be helpful to keep the reader connected to the importance of how this information pertains to driving specifically.

Regarding the list of top five manufacturers ...the list in the paper itself includes ten. This is initially a bit confusing.   Also, the author note ranking is done by the number of publications – a bit more information should be added here (i.e., what was the average number and or range, etc.). And why is this important to list in this manuscript?

The methods section indicates 113 articles were located in Medline and 59 in Embase.  Are these the only search places searched? And what was the time frame?  Where these systematically reviewed – any criteria for inclusion/exclusion?  This section needs more details.

Regarding inexperienced and experienced drivers – in line number 270 the authors compare inexperienced drivers with middle-aged drivers.  It seems they actually mean young drivers.  Thus, the section should reflect that one is comparing young, inexperienced drivers with middle-aged, experienced drivers (note: certainly there are inexperienced middle and older aged drivers. Do these older inexperienced drivers also process info differently compared to younger inexperienced drivers?)

Overall, the authors present a great deal of information on the importance of the visual system, ET technology, and related disease detection, but it is not entirely clear to me what the “results” are.  A description of a systematic review is not presented, nor are any actual results of this review.  

General edits

Line 43 – the sentence is awkward. Likely can remove “eye types” from the sentence.

Line 136 – delete space between cost and comma

Line 290 (and elsewhere) – older drivers – are these older as in 65 years and older? Please define.

Author Response

Ms. Frankie Yang, M.Sc.

Editor, MDPI

Room 2207, Jincheng Center, No. 21 Cuijingbeili, Tongzhou

District, Beijing 101121, China

Geriatrics Editorial Office

E-mail: geriatrics@mdpi.com

Manuscript ID: geriatrics-781345

Type of manuscript: Review

Title: The Role of Eye Tracking Technology in Assessing Older Driver Safety

Authors: David Carr *, Prateek Grover

Received: 6 April 2020

E-mails: dcarr@wustl.edu, groverp@wustl.edu

Submitted to section: Geriatric Neurology,

Aging and Driving: 2019

May 17, 2020

Dear Ms. Yang,

We have had the opportunity to review the excellent suggestions by the reviewers and have made all of their suggested changes without exception. We believe the manuscript is better organized and strengthened. We look forward to another review.

Sincerely,

David Carr

REVIEWER #2:

Regarding the list of top five manufacturers ...the list in the paper itself includes ten. This is initially a bit confusing.   Also, the author note ranking is done by the number of publications – a bit more information should be added here (i.e., what was the average number and or range, etc.). And why is this important to list in this manuscript?

We made some changes related to a similar comment by REVIEWER #1 above. We have added a statement that publications at least show buy in by the academic community and may give some assurances to those that are planning to embark in this area of study.

The methods section indicates 113 articles were located in Medline and 59 in Embase.  Are these the only search places searched? And what was the time frame?  Where these systematically reviewed – any criteria for inclusion/exclusion?  This section needs more details.

We realize the description to our search was vague. We have now moved the search methodology to the front of the manuscript with specific responses to these questions.

Regarding inexperienced and experienced drivers – in line number 270 the authors compare inexperienced drivers with middle-aged drivers.  It seems they actually mean young drivers.  Thus, the section should reflect that one is comparing young, inexperienced drivers with middle-aged, experienced drivers (note: certainly there are inexperienced middle and older aged drivers. Do these older inexperienced drivers also process info differently compared to younger inexperienced drivers?)

We have changed this section to state young drivers since inexperienced drivers could also include other age groups. A classic study by Brown et al in 1982 in Acid Anal Prev showed that inexperience regardless of age of onset of driving was associated with an increase crash risk compared to age matched controls. It is interesting to assume that there visual search patterns and the processing of information would be similar to young drivers. However, we are not aware of data on older inexperienced drivers (that starting driving for the first time in late life) in regards to eye tracking studies.

Overall, the authors present a great deal of information on the importance of the visual system, ET technology, and related disease detection, but it is not entirely clear to me what the “results” are.  A description of a systematic review is not presented, nor are any actual results of this review.  

See new Introduction and Methodology system that outlines the objectives and the search strategy.

General edits

Line 43 – the sentence is awkward. Likely can remove “eye types” from the sentence.

Removed.

Line 136 – delete space between cost and comma

Changed.

Line 290 (and elsewhere) – older drivers – are these older as in 65 years and older? Please define.

Yes. We have added this description to the beginning of this section.

Round 2

Reviewer 1 Report

minor suggestions:

Line 191: what type of presentation is likely to...

Line 202: delete space before the comma 

Line 257: clinical interviews and have not obtained...

line 266: coordination have been noted. 

line 267: ? with decreases in velocity and with... 

line 304: FTDbv? 

line 335: comma after "response"

line 475: inexperienced 

lines 488 to 490: some duplication could be eliminated 

Author Response

Review #1: Round Two

 Comments and Suggestions for Authors

These were somewhat difficult to find since the Line numbers cited below do not line up with the current manuscript version we were asked to upload. For instance, Line 191 suggestion was found on our current manuscript line 170. Would ask the Editors to review this to be sure we are all working from the same version. However, I believe we found them by simply subtracting about 20 lines…

Minor suggestions:

Line 191: what type of presentation is likely to...

Changed (new manuscript line 170)

Line 202: delete space before the comma 

Changed (new manuscript line 181)

Line 257: clinical interviews and have not obtained...

Changed. (new manuscript line 220).

line 266: coordination have been noted. 

Changed (new manuscript line 229).

line 267: ? with decreases in velocity and with... 

Changed. (new manuscript line 230).

line 304: FTDbv? 

Yes, fixed spelling and clarified with ( ). (new manuscript line 244)

line 335: comma after "response"

Changed. (new manuscript line 275).

line 475: inexperienced 

Was unsure where and what was being queried with this suggestion.

lines 488 to 490: some duplication could be eliminated 

Agree. We deleted a sentence on AD biomarkers that was redundant. (new line 419-420)

Reviewer 2 Report

The authors responded well to my concerns.  The manuscript is improved significantly. 

Author Response

Reviewer #2: Round Two

The authors responded well to my concerns.  The manuscript is improved significantly. 

Thank You.